# Rapid multiple-quantum three-dimensional fluorescence spectroscopy disentangles quantum pathways

Stefan Mueller[1], Julian Lüttig[1], Pavel Malý[1], Lei Ji[2], Jie Han[3], Michael Moos[4], Todd B. Marder[2], Uwe H.F. Bunz[5], Andreas Dreuw[3], Christoph Lambert[4,6] & Tobias Brixner [1,6]*

Coherent two-dimensional spectroscopy is a powerful tool for probing ultrafast quantum dynamics in complex systems. Several variants offer different types of information but typically require distinct beam geometries. Here we introduce population-based three-dimensional (3D) electronic spectroscopy and demonstrate the extraction of all fourth- and multiple sixth-order nonlinear signal contributions by employing 125-fold (1×5×5×5) phase cycling of a four-pulse sequence. Utilizing fluorescence detection and shot-to-shot pulse shaping in single-beam geometry, we obtain various 3D spectra of the dianion of TIPS-tetraazapentacene, a fluorophore with limited stability at ambient conditions. From this, we recover previously unknown characteristics of its electronic two-photon state. Rephasing and nonrephasing sixth-order contributions are measured without additional phasing that hampered previous attempts using noncollinear geometries. We systematically resolve all non-linear signals from the same dataset that can be acquired in 8 min. The approach is generalizable to other incoherent observables such as external photoelectrons, photo-currents, or photoions.

[1] Institut für Physikalische und Theoretische Chemie, Universität Würzburg, Am Hubland, 97074 Würzburg, Germany. [2] Institut für Anorganische Chemie and Institute for Sustainable Chemistry & Catalysis with Boron, Universität Würzburg, Am Hubland, 97074 Würzburg, Germany. [3] Interdisziplinäres Zentrum für Wissenschaftliches Rechnen und Physikalisch-Chemisches Institut, Ruprecht-Karls-Universität Heidelberg, Im Neuenheimer Feld 205, 69120 Heidelberg, Germany. [4] Institut für Organische Chemie, Universität Würzburg, Am Hubland, 97074 Würzburg, Germany. [5] Organisch-Chemisches Institut, Ruprecht-Karls-Universität Heidelberg, Im Neuenheimer Feld 270, 69120 Heidelberg, Germany. [6] Center for Nanosystems Chemistry (CNC), Universität Würzburg, Theodor-Boveri-Weg, 97074 Würzburg, Germany. *email: brixner@phys-chemie.uni-wuerzburg.de

nspired by multidimensional nuclear magnetic resonance (NMR) experiments that give extraordinary insight into chemical structure, optical multidimensional spectroscopic techniques reveal vibrational and electronic structure and dynamics employing sequences of ultrashort laser pulses. The enhanced dimensionality within these experiments enables the observation of cross peaks in the spectral domain. These may represent signatures of coupling phenomena, energy or charge transfer[1-4], photochemical reactions[5,6], vibrational or electronic coherences[7-13], or many-body interactions[14-17]. The great versatility of multidimensional techniques results from their ability to distinguish between specific nonlinear signal contributions. These contributions correspond to different subsets of optically driven quantum pathways through the excited-state manifold of a system.

In a two-dimensional (2D) experiment, a first laser pulse creates a coherence that evolves during the coherence time $\tau$. In one exemplary pathway, the interaction with a second pulse can lead to a population evolving during population time $T$, and the interaction with a third pulse to another coherence. The latter gives rise to a macroscopic polarization that emits a coherent electric field during the final time $t$. In the most common approach, this coherent field is heterodyne detected with a reference pulse to yield the amplitude-resolved and phase-resolved nonlinear optical signal. 2D Fourier transformation with respect to $\tau$ and $t$ results in a 2D spectrum as a function of frequencies $\omega_\tau$ and $\omega_t$, respectively, retaining a parametric dependence on $T$. This approach has been used extensively to investigate one-quantum (1Q) coherences, that is, coherent superpositions between states which oscillate at a frequency covered by the employed laser spectrum, and to decipher energy transfer mechanisms[18-21]. Broadband excitation in the visible may also generate superpositions of energetically close-lying states within one electronic manifold, also called zero-quantum (0Q) coherences, which are observed as oscillations as a function of $T$. In molecules, this often signifies nuclear wavepacket dynamics, oscillating on ground-state or excited-state electronic potential energy surfaces[7,22]. It is challenging to identify uniquely the origin of all oscillatory signals due to their spectral overlap. Using a third dimension may resolve some of the peaks[23-26]. Thus, one can distinguish vibrational coherence in ground and excited electronic states in molecules[22], unravel hidden vibrational couplings[27], or study ultrafast photochemistry[6]. Such 3D spectra may be used to isolate quantum coherence selectively, allowing for the possibility to reconstruct the Hamiltonian of a system[28,29].

Apart from higher dimensionality, higher orders of nonlinearity unveil further details that cannot directly be ascertained from lower-order signals. For instance, nonlinear signals that are of higher than third order in the interaction with the electric field enabled studying intricate details about electronic correlations in semiconductor nanostructures[16], tracking multistep energy transfer in light-harvesting complexes[30], and observing exciton–exciton annihilation in molecular aggregates[31]. While useful, it is challenging to capture all contributions with conventional noncollinear techniques because one must ensure phase stability between various optical beams. In addition, each signal contribution has to be measured separately by choosing the corresponding phase-matching condition, and scattering of excitation pulses as well as the nonresonant response from the environment leads to artifacts[32]. In this context, it was previously shown that a partially collinear pump–probe geometry offers many advantages due to its inherent phase stability, and the capability to isolate several, directly phased signals at once[33-37]. As another alternative that also preserves these advantages, one can employ phase modulation or phase cycling with fluorescence detection[38-47]. Fluorescence detection reveals features

that would be hidden with coherent detection[47,48]. Another key feature of fluorescence-based detection is the excellent sensitivity for higher-order signal contributions[49], since scattered excitation light does not contribute. Most fluorescence-based multidimensional experiments focus on fourth-order nonlinear signals, while studies of high-order population-based spectroscopy are scarce[49-51]. Apart from that, it is desirable to design a versatile and convenient approach to gain maximum nonlinear information content from a single data set in order to investigate systems with limited (photo) chemical stability without the need to carry out separate multidimensional experiments for each signal contribution. Such an approach should offer direct comparability between different signal contributions that is beneficial for interpretation purposes, especially because nonlinear signals show a strong dependence on the employed excitation conditions that might not be identical throughout separate measurements.

In this work, we address the aspect of gaining multiple nonlinear signals from a single rapid measurement and demonstrate this by recording a variety of multiple-quantum, fourth- and sixth-order 3D spectra simultaneously. This allows us to extract certain sets out of a vast manifold of quantum pathways where each set provides rich and selective information about correlations between excited states, vibrational coherences, and excited-state dynamics. We show how to obtain the information systematically from only one raw data set by appropriate weighting with Liouville pathway-specific phase factors. Thus we can assign 0Q, 1Q, and two-quantum (2Q) coherence dynamics to the three conjugate time delays employed in the experiment. For demonstration, we choose the dianion of the bis(triisopropylsilyl)ethynyl-substituted symmetric tetraazapentacene (TIPS–TAP$^{2-}$)[52,53], dissolved in tetrahydrofuran (THF), which nicely illustrates the application potential of the technique because the highly reduced compound is extremely sensitive toward traces of oxygen and moisture, thus having limited stability at ambient conditions, and all data from two different orders of nonlinearity are obtained within only a few minutes of measurement time. The TIPS–TAP compound and related species are of considerable interest for use in organic field effect transistors (OFETs) due to their ability to serve as efficient negative charge carriers. In addition, and of specific relevance to this study, the TIPS–TAP dianion, formed by two-electron reduction of the neutral compound, has a remarkably high fluorescence quantum yield of 95% in Et$_2$O solution[52,53].

## Results
**Method development.** We employ a single-beam geometry and thus cannot use phase matching to distinguish between various signal contributions. Instead, we employ phase cycling (Fig. 1)[36-38,40,42,54,55]. The key idea is to vary the phases of the excitation pulses with step sizes fine enough to resolve different signal contributions without aliasing. Because the laser is not carrier-envelope phase-stabilized, it is adequate to operate with interpulse phases $\Delta\varphi_{i1}$ ($i = 1, 2, 3$) that are cycled in discrete steps between 0 and $2\pi$ with a pulse shaper (DAZZLER, Fastlite). Thus, the raw data matrix $p$ obtained from the experiment represents integrated fluorescence intensities as a function of sampled time delays and interpulse phases. Various phase-cycling schemes for fourth-order population-based 2D spectroscopy were discussed in the literature, aiming to preserve an experimentally manageable amount of phase steps[54]. Phase-cycling protocols employing an increased number of phase steps generate a denser grid in phase space resolving more nonlinear signal contributions within a single experiment. The associated significant increase in acquisition time can be compensated by our rapid-scan approach[55]. In our experiment, we

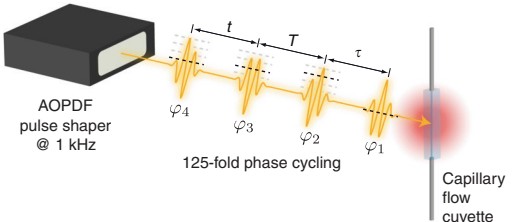

**Fig. 1** Scheme of the experiment. An acousto-optic programmable dispersive filter (AOPDF) creates a phase-coherent pulse sequence with individual pulse phases $\varphi_i$ ($i = 1, 2, 3, 4$) that is employed in the multidimensional experiment. Besides scanning the interpulse delays $\tau$, $T$, and $t$, we further sample the phases of the second, third, and fourth pulse five times each and finally detect the integrated fluorescence intensity from the sample as a function of these excitation pulse sequence parameters

sampled, in addition to the time delays $\tau$, $T$, and $t$ between the pulses, the interpulse phases by means of a 125-fold ($1 \times 5 \times 5 \times 5$) phase-cycling scheme (Fig. 1), where the phases of the second, third, and fourth pulse were sampled in five steps each. A detailed schematic of the experimental setup is given in Supplementary Fig. 1 and described in Supplementary Note 1.

After the experiment, the raw data $p$ is decoded by weighting it with phase factors which recover complex-valued nonlinear signal contributions $\tilde{p}$ by using

$$\tilde{p}(\tau, T, t, \beta, \gamma, \delta) = \frac{1}{125} \sum_{l=0}^{4} \sum_{m=0}^{4} \sum_{n=0}^{4} p(\tau, T, t, l\Delta\varphi_{21}, m\Delta\varphi_{31}, n\Delta\varphi_{41})$$
$$\times \exp(-il\beta\Delta\varphi_{21}) \exp(-im\gamma\Delta\varphi_{31}) \exp(-in\delta\Delta\varphi_{41}),$$

(1)

where $\Delta\varphi_{ij}$ denotes interpulse phases between pulses $i$ and $j$, and $\beta$, $\gamma$, and $\delta$ are the weighting parameters[54]. According to phase-cycling theory and double-sided Feynman diagram analysis (see Supplementary Note 2)[54,56], we identified that this scheme is able to resolve and distinguish between three fourth-order and twelve sixth-order contributions (Table 1), all of them simultaneously, i.e., from the same raw data set.

With 125-fold phase cycling, five-step phase sampling ensures discrimination between integer values ranging from $-2$ to $+2$ for $\beta$, $\gamma$, and $\delta$ so that all listed fourth-order and sixth-order contributions are obtained simultaneously, but can be resolved individually. Besides the fourth-order contributions, the selection of the sixth-order contributions is dictated by choosing the weights of $\beta$, $\gamma$, and $\delta$ themselves because at the moment one of the factors is set to $\pm 2$, the weighting will uniquely pick a contribution which originates from a pulse sequence in which two of the four pulses interact twice, resulting in six interactions in total. Performing 3D Fourier transforms of each contribution with respect to $\tau$, $T$, and $t$ generates 3D spectra as will be shown below using our experimental data. All contributions are inherently phased correctly with respect to each other. This is an advantage compared with noncollinear coherence-detected multidimensional spectroscopy, for which one must usually perform an auxiliary procedure, such as an additional pump–probe experiment, in order to assign the correct phase information[57]. However, analogously to coherently detected experiments[31], if the excitation density is high enough to generate sixth-order signals, these may also contaminate the fourth-order ones. Hence, whenever we use the term fourth-order signal, we mean a nonlinear signal that is predominantly of fourth order (see Supplementary Note 3). Apart from that, an advantage of our technique is that we are able to simultaneously isolate the higher-order contribution by appropriate phase cycling.

We sampled every temporal dimension in steps of 6 fs utilizing the fully rotating frame of the pulse shaper's reference frequency

## Table 1 Summary of fourth-order and sixth-order nonlinear signal contributions

**Fourth-order contributions $\tilde{p}^{(4)}$**

| | $\tau$ | $T$ | $t$ | $\beta$ | $\gamma$ | $\delta$ |
|---|---|---|---|---|---|---|
| R[a] | 1Q | 0Q | 1Q | +1 | +1 | −1 |
| NR | 1Q | 0Q | 1Q | −1 | +1 | −1 |
| NR | 1Q | 2Q | 1Q | +1 | −1 | −1 |

**Sixth-order contributions $\tilde{p}^{(6)}$**

| | $\tau$ | $T$ | $t$ | $\beta$ | $\gamma$ | $\delta$ |
|---|---|---|---|---|---|---|
| R[b,c] | 1Q | 1Q | 1Q | +2 | −2 | +1 |
| R[a] | 2Q | 0Q | 1Q | +2 | +1 | −1 |
| NR | 2Q | 0Q | 1Q | −2 | +1 | −1 |
| R[a] | 1Q | 0Q | 2Q | +1 | +2 | −2 |
| NR | 1Q | 0Q | 2Q | −1 | +2 | −2 |
| R[a,c] | 2Q | 1Q | 1Q | +1 | +2 | −1 |
| R[a,b] | 1Q | 1Q | 2Q | +2 | +1 | −2 |
| NR | 2Q | 1Q | 2Q | −1 | +1 | −2 |
| NR | 1Q | 3Q | 1Q | +2 | −2 | −1 |
| NR | 2Q | 3Q | 1Q | +1 | −2 | −1 |
| NR | 1Q | 3Q | 2Q | +2 | −1 | −2 |
| NR | 2Q | 3Q | 2Q | +1 | −1 | −2 |

The signal contributions listed here can be resolved and isolated from a 125-fold ($1 \times 5 \times 5 \times 5$) phase-cycling scheme using incoherent detection. The first four columns on the left give the label of the nonlinear signal contribution that probes specific types of quantum coherence over the three temporal dimensions $\tau$, $T$, and $t$. Their respective weight factors $\beta$, $\gamma$, and $\delta$ are listed on the right. All contributions can further be classified as either fully nonrephasing (NR, no phase conjugation between coherences) or rephasing (R), whereas for the latter the superscripts denote which particular dimensions hold the phase conjugation that is introduced during a rephasing pulse sequence; a: between $\tau$ and $t$, b: between $\tau$ and $T$, c: between $T$ and $t$

of 508.1 THz (2.1 eV) in order to find a compromise regarding acquisition time with respect to the limited sample stability. In total, $15 \times 15 \times 15 \times 125 = 421{,}875$ different pulse trains were streamed in order to acquire a full 3D data set. At the 1 kHz repetition rate of our setup, the pure acquisition time for the full multidimensional data set is only 8 min (without averaging). In our sampling protocol, we first sample through all phase-cycling steps for fixed interpulse delays to prevent errors in reconstructing the phase-sensitive nonlinear signal contributions by Eq. (1) (for further details see Supplementary Note 1). With the broad spectral bandwidth (70 nm) of the employed laser pulses, we resolve high-frequency vibrational coherence in the 0Q domain, 1Q coherence between ground- and singly-excited electronic states as well as between singly- and higher-excited states, and 2Q coherences between ground- and higher-excited electronic states via multiphoton processes. Li et al. previously introduced multiphoton 2D fluorescence spectroscopy, where signatures of one-photon and two-photon transitions into the same electronic state could be obtained[58]. In this work, however, we will consider one-photon and two-photon transitions into different electronic states. Probing 3Q coherences would lead to aliasing artifacts in the present example. Therefore, we will restrict the discussion below to the signal contributions that did not involve any 3Q coherence. However, using more temporal sampling steps would allow the unique recovery of 3Q contributions as well with the same setup, again within one unified measurement protocol.

Our technique is currently limited to a maximum temporal shaping window of 8 ps for every pulse sequence according to the length of the chosen AOPDF crystal. At such delays, one can measure electronic and vibrational coherence dynamics of dissolved molecular systems that proceed on time scales up to several picoseconds. In cases of much slower dephasing, such as in some semiconductor nanostructures[15,16,59], measurements of all relevant dynamics becomes challenging. Another example of slow dynamics proceeding at the 10 to 100 ps time scale is exciton diffusion dynamics in some molecular aggregates[31]. Nevertheless,

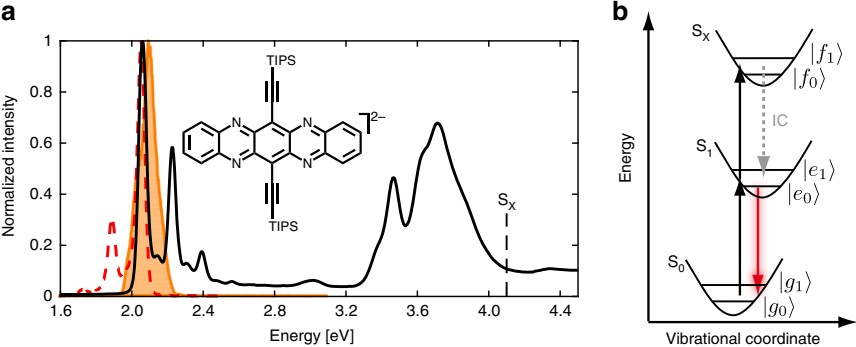

**Fig. 2** Sample spectra and energy scheme. **a** Linear absorption (black, determined via spectroelectrochemistry in THF/N($n$-Bu)$_4$PF$_6$ of TIPS–TAP$^{2-}$ (molecular structure shown in the inset). The laser spectrum employed in the multidimensional experiment is displayed as an orange shaded area. The emission spectrum of TIPS–TAP$^{2-}$ in THF, which was acquired immediately before the multidimensional experiment, is shown as a red dashed line. Absorption and emission spectra are normalized to their respective maximum value. The black dashed line at 4.1 eV marks the energy of the doubly excited state $S_x$. **b** Energy-level scheme used for interpreting the 3D spectra with the dashed gray arrow denoting internal conversion (IC) and the red arrow the observed fluorescence

because of the absence of spurious coherent artifacts for early population times, our technique has great application potential to study ultrafast processes such as rapid exciton–exciton annihilation in many classes of molecular aggregates, and the single-beam geometry provides a compact, stable, and convenient access.

**Fourth-order 3D spectra**. The linear absorption[53] and fluorescence spectra of TIPS–TAP$^{2-}$ are shown in Fig. 2a together with the laser spectrum. The dianion features well-resolved peaks in both absorption and fluorescence spectra together with a small Stokes shift. The main band of the $S_0 \rightarrow S_1$ $\pi$–$\pi^\star$ transition[52] at 2.06 eV is well covered by the laser spectrum, having further very small amplitude in the region of the main vibrational progression band, which is located at 2.23 eV. This progression is assigned to a $\approx$1400 cm$^{-1}$ ($\approx$0.17 eV) vibrational mode of the conjugated system[53]. Based on that and on experimental observations, we construct an energy-level scheme depicted in Fig. 2b, incorporating the single 0.17 eV mode being coupled to the electronic singlet states $S_0$, $S_1$ as well as to a highly excited two-photon singlet state $S_x$ with different displacements along the vibrational coordinate (see also below). For the following, we use the labels $|g_v\rangle$, $|e_v\rangle$, and $|f_v\rangle$ to denote electronic ground-, first excited, and doubly excited states, respectively, where the subscript $v$ denotes the quantum of vibrational excitation.

In Fig. 3a (left), we present the experimental absolute-valued rephasing 1Q–0Q–1Q 3D spectrum, which we extracted from the raw data by using the weights from Table 1 according to its phase signature[54] of $\varphi_{R1Q0Q1Q} = -\varphi_1 + \varphi_2 + \varphi_3 - \varphi_4$. Throughout this section, we display the 3D spectral solids with three isosurfaces of varying transparency, visualizing the signal amplitude. In addition, we project the solid onto 2D frequency planes, which are represented as contour plots that are normalized to the highest absolute value each. Thus, different types of 2D spectra can be extracted as well. In case of the rephasing 1Q–0Q–1Q spectrum, the 2D projections deliver the photon-echo (1Q–1Q) spectrum for $T = 0$ on the bottom ($\hbar\omega_\tau$, $\hbar\omega_t$) plane, the 1Q–0Q spectrum at $t = 0$ on the ($\hbar\omega_\tau$, $\Delta\hbar\omega_T$) plane, and the 0Q–1Q spectrum at $\tau = 0$ on the ($\hbar\omega_t$, $\Delta\hbar\omega_T$) plane. The 3D solid displays a dominating feature centered around a coordinate of ($\hbar\omega_\tau$, $\Delta\hbar\omega_T$, $\hbar\omega_t$) = (2.06, 0, 2.06) eV that exhibits a characteristic star-shaped lineshape[29,60]. This peak coordinate corresponds exactly to the maximum of the linear absorption spectrum, which is attributed to the transition from state $|g_0\rangle$ to $|e_0\rangle$ in the model (Fig. 2b). It is accompanied by a much weaker feature at (2.23, 0, 2.23) eV, which corresponds to the transition from $|g_0\rangle$ to $|e_1\rangle$.

The overall amplitude of the 3D solid is determined by the multiplication of $|e_0\rangle$ and $|e_1\rangle$ with the laser spectrum. The solid is elongated toward higher 1Q energies which reflects coupling to the progression band $|e_1\rangle$ at 2.23 eV, mediated by the common ground state $|g_0\rangle$. Mediation by a hot ground state $|g_1\rangle$ can be excluded because this would involve an $|e_0\rangle \rightarrow |g_1\rangle$ transition with a transition energy of 1.89 eV, where the laser spectrum has no intensity. As further shown in Supplementary Note 5 and Supplementary Fig. 7 and discussed in a similar scenario[61], the filtering effect by the laser spectrum removes from the complete manifold a number of pathways that involve the transition between $|e_0\rangle$ and $|g_0\rangle$, also in the nonrephasing 1Q–0Q–1Q contribution. This simplifies the interpretation.

Apart from these features being located within the $\Delta\hbar\omega_T = 0$ plane, which reflects the fact that the system evolves in a population state during $T$, we observe two additional weak cross peaks A and B. In order to separate them from the dominating peaks, we isolate the voxels within the spectral volumes that span the regions in the vicinity of peak A and B, which is indicated by red and blue cuboids, respectively (Fig. 3a, right). By determining the coordinates of the maximum amplitude of each isolated cross-peak, we find that these are shifted along the $\Delta\hbar\omega_T$ axis by +188 meV (A) and −172 meV (B). In view of the effective spectral resolution (15.7 meV), these values are in fair agreement with the energy spacing $\hbar\omega_{0Q} \approx \pm$ 0.17 eV between vibrational levels within $S_0$ and $S_1$, and thus they originate from vibrational coherence. These cross peaks can be described by double-sided Feynman diagrams[56] (Fig. 3b), where the wavy arrows pointing to the right (left) denote the laser fields, interacting with a phase of $+\varphi$ ($-\varphi$). The sign of the 0Q coherence frequency determines whether the coherence oscillates positively, e.g., $|e_1\rangle\langle e_0| \propto \exp(-i\omega_{0Q}T)$, or negatively with $|e_0\rangle\langle e_1| \propto \exp(+i\omega_{0Q}T)$[29,61]. As previously demonstrated, analyzing these oppositely signed signatures has proven to be a powerful tool to decipher the origin of the underlying quantum coherences[62,63]. Peak A is described by a single Feynman diagram involving vibrational coherence in the first excited electronic state that oscillates with a positive frequency. In contrast, Peak B can be described by two diagrams that include vibrational wavepacket evolutions oscillating with a negative frequency within both the ground and the first excited electronic state. There is no way to construct a diagram in which a positively oscillating $|g_1\rangle\langle g_0|$ coherence is created, except if one assumes that the system is initially in a vibrationally excited electronic ground state which is unlikely because the amount of thermal energy that was present

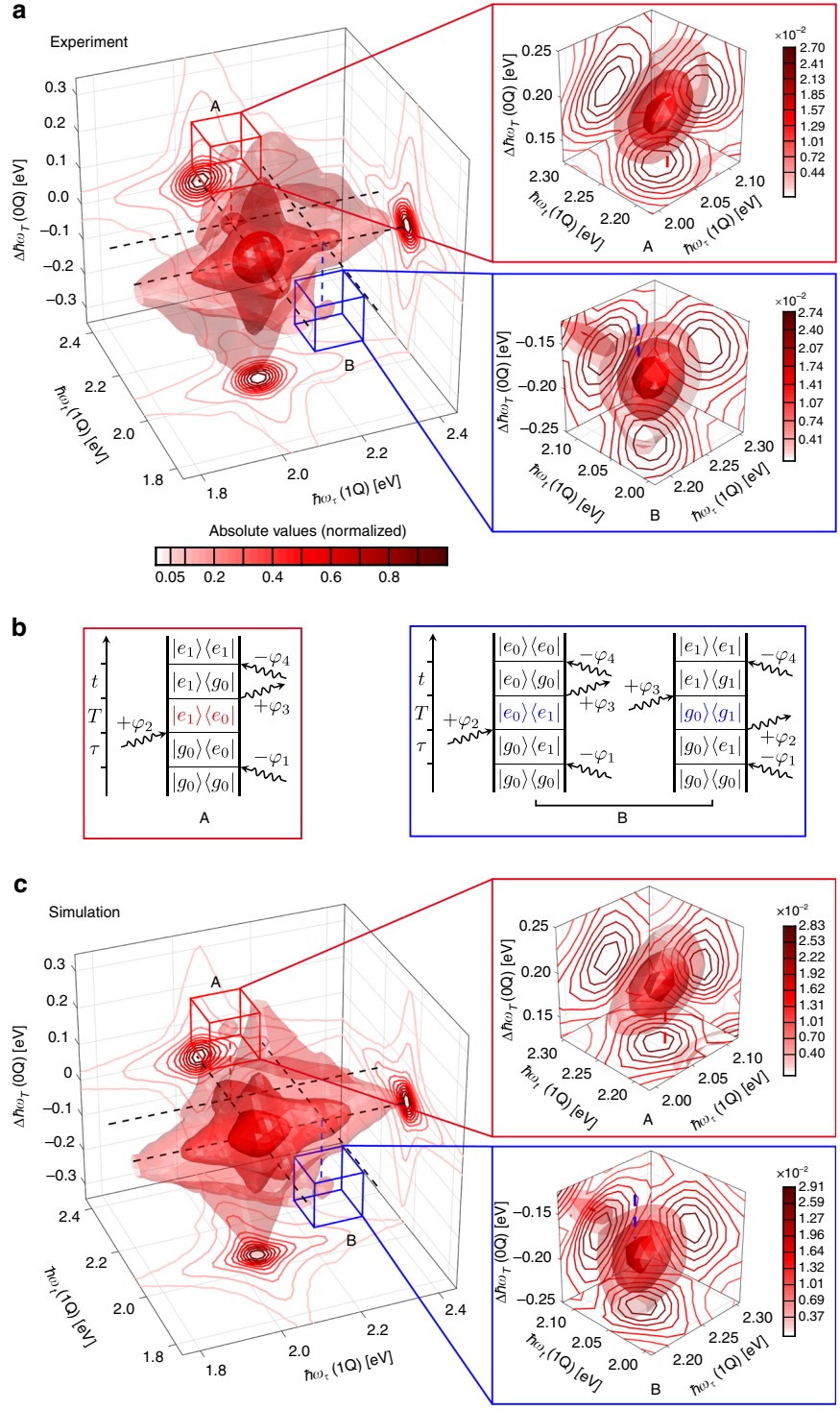

**Fig. 3** Fourth-order rephasing 1Q–0Q–1Q 3D spectrum of TIPS–TAP$^{2-}$ in THF. **a** Experimental and **c** simulated rephasing 1Q–0Q–1Q 3D spectra, shown in absolute values. The energies of the main absorption band at 2.06 eV and the progression band at 2.23 eV are indicated as dashed black lines in the $\Delta\hbar\omega_T = 0$ plane as a guide for the eye. Red and blue dashed vertical lines mark positive and negative shifts of 0.17 eV, respectively, along the 0Q axis $\Delta\hbar\omega_T$. Isosurfaces are drawn at 2.3, 9.0, and 35% of the maximal signal amplitude. Contour lines of the 2D projections of the spectral solids are drawn at linearly spaced levels of the normalized signal amplitude with an additional contour line at 0.05. The spectral volumes inside the red and blue cuboids, which are centered around peaks A and B are isolated and shown on the right. There, isosurfaces are drawn at 50, 70, and 90% of the isolated maximal signal amplitudes whereas the color bars indicate the signal levels (normalized to the highest absolute value of the respective full 3D spectrum) for each contour line. **b** Double-sided Feynman diagrams that describe the peaks A (red) and B (blue)

at the experiment ($\approx$26 meV) is much less than the 0Q energy ($\approx$170 meV). Interestingly, although two pathways could contribute to peak B in principle, its amplitude is nearly identical to peak A (Fig. 3a, right), indicating that only one of the two pathways for peak B contributes significantly. Taking into account the Huang–Rhys factor of the $\approx$0.17 eV mode, which we estimated to be $S_{HR} \approx 0.6$ from linear absorption, leads to a low transition strength for the $|g_1\rangle \rightarrow |e_1\rangle$ transition that is involved twice in the right B pathway in Fig. 3b, strongly diminishing its contribution to the peak. Thus, while peak A represents an isolated single Liouville pathway, peak B predominantly bears the signature of the excited-state vibrational coherence over $T$. As evident from Fig. 3, the experimental results (Fig. 3a) can be very well reproduced by simulations (Fig. 3c) using Lindblad theory[64], in terms of peak positions as well as their shapes and amplitudes, even for the peaks A and B (for details about the simulation model see below and Supplementary Note 6). Overall, this successfully demonstrates the decongestion that 3D spectroscopy offers, yielding a separation of population and wavepacket contributions in the studied molecule which in turn enables us to isolate signatures of vibrational coherence in the excited electronic state and distinguish them from vibrations in the electronic ground state.

While 1Q–0Q–1Q spectra give extensive information about the manifold of singly excited states, it is difficult to reveal direct signatures involving higher excited states which could be accessed via excited-state absorption (ESA). This issue is a complication for interpreting any type of third-order spectroscopy, be it pump–probe transient absorption or conventional electronic 2D spectroscopy, i.e., it is challenging to assess the influence of higher excited states on the observed transient spectra because, typically, not much is known about them. Thus, it is desirable to design nonlinear spectroscopy experiments that specifically address these higher lying states. We will turn to such experiments now and implement 1Q–2Q–1Q 3D spectroscopy. The results obtained therein will also justify why, while discussing the 1Q–0Q–1Q spectra, we neglected any ESA signatures that could potentially contribute to the 1Q–0Q–1Q signals. (However, we already incorporated the higher excited state into the simulation model, but the 1Q–0Q–1Q ESA pathways cancel in pairs, as also discussed in Supplementary Note 4).

In order to probe highly excited states, it is necessary to prepare them as a 2Q coherence via an interaction phase $\pm 2\varphi$ by a single laser pulse[54,65] or by $\pm\varphi$ from two pulses each[66]. Concerning the latter, applying the weights that correspond to the phase signature $\varphi_{1Q2Q1Q} = +\varphi_1 + \varphi_2 - \varphi_3 - \varphi_4$ (Table 1) extracts all nonrephasing fourth-order pathways in which a 2Q coherence evolves during $T$ and that can be visualized as the experimental 1Q–2Q–1Q 3D spectrum (Fig. 4a).

Now, the $\hbar\omega_T$ axis contains information about electronic states that lie within two quanta of the excitation laser spectrum so that one can probe how these states are coupled to the manifold of singly excited states. The 3D signal peaks at 4.10 eV along the 2Q axis, indicating the presence of a higher-excited electronic state $S_x$. The solid shows further a vibrational shoulder along $\hbar\omega_T$ with a higher magnitude compared with the 1Q progression band, resulting from the multiplication with the laser spectrum which has its maximum at $\approx$4.20 eV in the 2Q domain. The linear absorption spectrum in the UV range (see Fig. 2a) does not exhibit a significant absorption band at 4.10 eV, suggesting that the probed 2Q state is a two-photon allowed (but one-photon forbidden) state, which is reasonable considering the inversion symmetry of the molecule. Time-dependent density–functional theory calculations (TD–DFT) also support such a two-photon state which can be attributed to $S_{74}$ and that is accessed by two consecutive nearly degenerate one-photon transitions $S_0 \rightarrow S_1$

and $S_1 \rightarrow S_{74}$, respectively, both having high oscillator strengths, whereas the direct $S_0 \rightarrow S_{74}$ transition has zero oscillator strength (see Supplementary Note 7 and Supplementary Figs. 8 and 9). We sought a separate experimental signature of the 2Q state by performing a two-photon absorption (2PA) measurement. The strong one-photon absorption, however, superimposes any potential 2PA signal so that it is impossible to extract the 2PA spectrum of this compound by conventional 2PA techniques. Nevertheless, this underlines the significant advantage of the use of phase cycling and two-quantum 3D spectroscopy, because here the desired nonlinear signal is isolated unambiguously by its unique phase signature.

The 1Q–2Q–1Q spectral solid can be projected onto the three 2D planes to extract a nonrephasing 1Q–1Q spectrum at $T = 0$ along ($\hbar\omega_\tau$, $\hbar\omega_t$), the 2Q–1Q spectrum along ($\hbar\omega_t$, $\hbar\omega_T$), and the 1Q–2Q spectrum along ($\hbar\omega_\tau$, $\hbar\omega_T$)[23]. Fluorescence-detected 2Q spectroscopy is sensitive to relaxation channels of highly excited states because all relaxation processes that happen after the perturbation by the light fields until the final emission of fluorescence in fact matter, which is in contrast to coherently detected multidimensional spectroscopy[48,67]. In that sense, a cross check between the 2Q–1Q and 1Q–2Q 2D spectra offers the ability to draw conclusions about the relaxation pathways of the 2Q manifold[65]. From Fig. 4a, it is apparent that these 2D projections are nearly identical in relative signal amplitude, which is an indication of highly efficient internal conversion from $S_x$ to $S_1$ as expected for a system with such a high fluorescence quantum yield (95% in Et₂O[52,53]). The double-sided Feynman diagrams in Fig. 4b explain that the equality of both projections results from a total cancellation of the opposite-signed excited-state absorption pathways $Q_B$ and $Q_C$. Due to a complete nonradiative population transfer from $|f_v\rangle\langle f_v|$ to $|e_v\rangle\langle e_v|$, the pathway-specific quantum yield[68] $\Phi_f$ is equal to $\Phi_e$. This erases any signal stemming from $|f_v\rangle\langle e_v|$ coherences so that both 2D projections only feature $|e_v\rangle\langle g_0|$ coherence signals along their respective 1Q axes. Contributions from $|f_v\rangle\langle e_v|$ coherences would lead to an asymmetry of the 3D solid. Consequently, the whole 3D spectrum can be described by pathway type $Q_A$ only. Thus, the 1Q–2Q–1Q 3D spectrum delivers detailed information on the highly excited 2Q state and its relaxation dynamics.

The simulated 1Q–2Q–1Q spectrum (Fig. 4c) reproduces well the overall symmetric shape of the solid, reflecting identical 2Q–1Q and 1Q–2Q projections, where we estimated an internal conversion time constant of 100 fs. In the simulation model, we derived the transition strengths into the $S_x$ state with the aid of TD–DFT calculations (see Supplementary Note 6). In addition, by performing simulations with varying Huang–Rhys factor of the $\approx$0.17 eV mode between $S_1$ and $S_x$, we found that a $S_{HR}$ of ~0.05 best reproduces the experiment (especially the lineshape along the 2Q axis). This reflects a much smaller displacement between the two-photon and the first excited state compared with that between first excited state and ground state. Furthermore, a pure dephasing time for all $|f_v\rangle\langle e_v|$ coherences of 300 fs yields good reproduction, which is much higher than those of the $|e_v\rangle\langle g_v|$ coherences (90 fs), thus indicating a lower transition energy fluctuation between $S_1$ and $S_x$ compared with $S_0$ and $S_1$. It is also worth noting that in coherently detected 2Q spectroscopy, when $|e_v\rangle\langle g_v|$ and $|f_v\rangle\langle e_v|$ coherence energies are nearly degenerate as in the present system, the signal would vanish due to destructive interference of the involved pathways[69]. In contrast, the 2Q-associated pathways in population-based detection differ due to the additional order in perturbation, so that the signal is preserved by a single pathway type[68]. In the present case, this is an advantage because one can then analyze the signal, whereas the vanishing signal of coherently detected 2Q spectra obviously could not be analyzed. To conclude, these results demonstrate that our approach provides rich information about

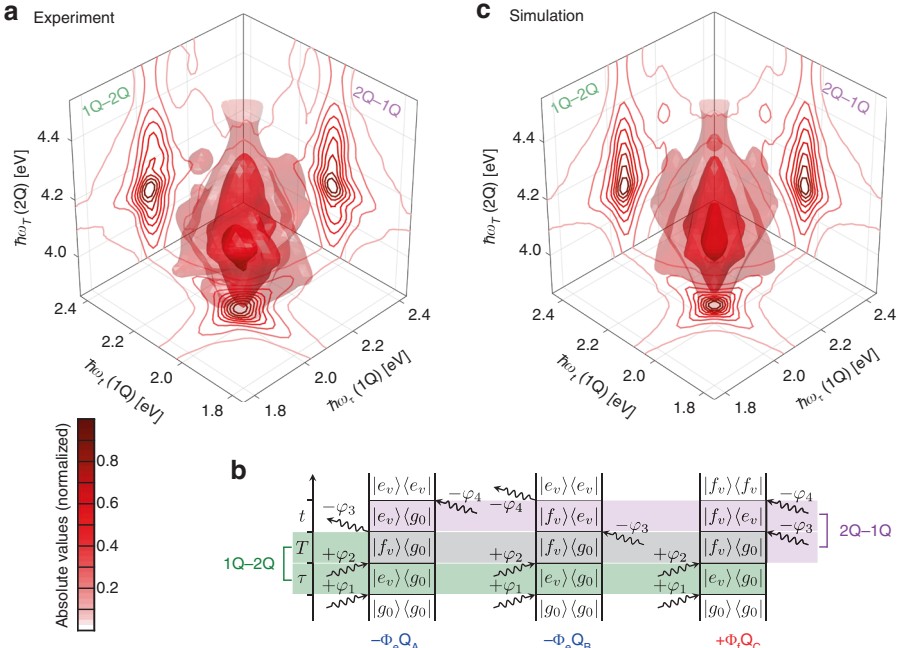

**Fig. 4** Fourth-order three-dimensional 1Q–2Q-1Q spectrum of TIPS–TAP$^{2-}$ in THF. **a** Experimental and **c** simulated 3D spectra, both shown in absolute values. Isosurfaces are drawn at 11, 25, and 60% of the maximal signal. Contour lines of the 2D projections are drawn at linearly spaced levels of each normalized projection. **b** Double-sided Feynman diagrams that correspond to the 1Q–2Q–1Q process, highlighting the coherence density matrix elements that are relevant for 1Q–2Q and 2Q–1Q 2D spectra in green and pink, respectively. Each pathway is labeled according to its sign, where Φ denotes the pathway-specific fluorescence quantum yield that is dependent on the finally prepared population state (marked by the index)

the electronic structure of the system that can otherwise be obscured in other methods.

**Sixth-order 3D spectra.** While the previous discussion involved responses from fourth-order nonlinear processes, in which each pulse interacts once with the system, the employed phase cycling also selectively captures processes in which six light-field interactions are distributed over four pulses. For the following, we want to focus on sixth-order processes that probe 0Q, 1Q, and 2Q coherences simultaneously, such as the rephasing 2Q–0Q–1Q contribution. Compared with the fourth-order 1Q–2Q–1Q process, in a sixth-order process the generation of the 2Q coherence results from a double interaction of a single laser pulse. The overall six-fold interaction pattern allows us to rephase a 2Q coherence by means of a 1Q coherence, thus the homogeneous linewidth of both 1Q and 2Q coherences contribute to the spectral response. Therefore, this sixth-order sequence is analogous to the coherently detected fifth-order rephasing 2Q sequence, which was first designed by Fulmer, Ding, and Zanni for 2D IR spectroscopy[70,71], but here carried out using fluorescence. There are also other coherently detected signal variants that employ this process in the optical regime[31,59]. However, extending the earlier work we are able to isolate both rephasing and nonrephasing 2Q–0Q–1Q as well as 1Q–0Q–2Q contributions directly in amplitude and phase without the need of additional phasing. Analogous to the common procedure for lower-order 2D spectra, adding the real parts of the properly rotated rephasing and nonrephasing 2Q–0Q–1Q contributions with phase signatures $\varphi_{R2Q0Q1Q} = -2\varphi_1 + 2\varphi_2 + \varphi_3 - \varphi_4$ and $\varphi_{NR2Q0Q1Q} = 2\varphi_1 - 2\varphi_2 + \varphi_3 - \varphi_4$, respectively, enables us to receive purely absorptive 2Q–1Q spectra. The 2Q–1Q spectrum at $T = 0$ can be extracted by taking the bottom projection of the 3D spectrum in Fig. 5a.

As is typical for a higher-order nonlinear signal, the overall signal amplitude is much lower than in the fourth-order spectra, which manifests as increased noise compared with the fourth-order spectra. Still, the signal quality is quite good due to the use

of fluorescence as an observable that excludes artifacts from nonresonant solvent contributions and scattered excitation light, and also to the shot-to-shot rapid scanning implementation. Thus, even this sixth-order signal shows a remarkable agreement with the theoretical prediction (Fig. 5b).

We observe features at positions A and C signifying that $|e_0\rangle$ and $|e_1\rangle$ are coupled to the lowest level of the doubly excited manifold, $|f_0\rangle$, whereas features B and D indicate that the singly excited manifold is coupled to $|f_1\rangle$ as well. This leads to a rectangular arrangement of features in the $\Delta\hbar\omega_T = 0$ plane which is also well reproduced by the simulation (Fig. 5b), further verifying the absence of lower-order cascaded contributions. Additional unwanted signals, such as those originating from many-particle excitations of noninteracting molecules[72] are not present in our experiment (for a detailed discussion see Supplementary Note 9). Another source for unwanted signals is so-called incoherent population mixing, that is, signals arising from interacting excited-state populations, as discussed by Grégoire et al., which was observed in solid-state semiconductors under high excitation densities[73]. In our experiment, we determined the number of absorbed photons per molecule[31] to be 0.26, which would in principle facilitate the generation of such incoherent signals by bimolecular processes. Considering the relevance of this value for multiparticle interactions, however, we have to take into account the relative distance of chromophores because for systems that do not form aggregates (such as in the present case), the probability of bimolecular processes is generally low. According to the suggestion of Tiwari et al., we have estimated the effect of multiparticle interactions for the present sample using Förster resonance energy transfer (FRET) as a crucial step that would facilitate the interaction of populations of two distinct molecules by subsequent exciton–exciton annihilation[43]. We calculated the Förster critical concentration[74,75] for this effect to be 0.54 mM, which is above our concentration

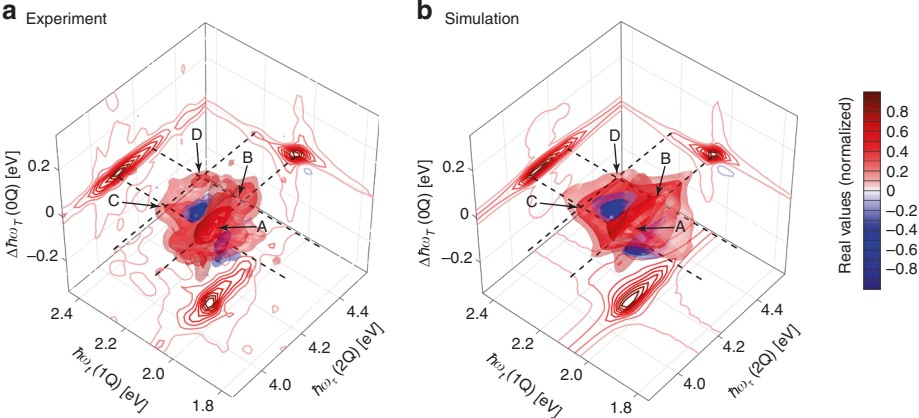

**a** Experiment  **b** Simulation

**Fig. 5** Sixth-order 2Q–0Q–1Q three-dimensional (3D) spectrum (real part) of TIPS–TAP$^{2-}$ in THF. The 3D spectra shown are obtained by the sum of the rephasing and nonrephasing 2Q–0Q–1Q contributions. **a** Experimental and **b** simulated 3D spectra. Isosurfaces are drawn at −12, −10, −7, 7, 11, and 35% of the normalized signal amplitude. Contour lines are drawn at 0.1 steps as indicated by the color bar. The bottom projections of the solids show the purely absorptive 2Q–1Q 2D spectrum at $T = 0$ fs. The dashed lines in the $\Delta\hbar\omega_T = 0$ plane denote the energies of the main absorption band at 2.06 eV and the progression band at 2.23 eV along $\hbar\omega_t$ and the energies of the 2Q levels at 4.10 and 4.27 eV along $\hbar\omega_\tau$. The letters A, B, C, D mark peaks, which are discussed in the article

(0.40 mM) so that we expect such contributions in our experiment not to be present.

Besides the inverted sign in the 3D solid compared with fourth-order contributions (see also Supplementary Note 8), strong evidence for the isolation of sixth-order signals is presented in Fig. 6, where we compare transients over $T$ of the rephasing fourth-order 1Q–0Q–1Q (Fig. 6a) and the rephasing sixth-order 2Q–0Q–1Q contributions (Fig. 6b), taken at above-diagonal cross-peak positions. The respective Liouville pathways (Fig. 6c, d) that contain these 0Q coherences only differ in the type of electronic quantum coherence that evolves during $\tau$, but the coherence evolution during $T$ is the same (marked in blue and red, respectively). The 0Q oscillation period ($\approx 24$ fs) corresponds well to the 0.17 eV vibrational mode in both cases. In principle, sixth-order 2Q–0Q–1Q processes are capable of tracking 0Q coherence dynamics within higher-excited electronic states. This is impossible to see with lower-order spectroscopic techniques. However, we attribute the present oscillation to 0Q coherence within the $S_1$ state because all pathways that bear possible 0Q coherence in $S_x$ cancel due to internal conversion (see Supplementary Note 10 and Supplementary Fig. 12). Importantly, we observe a phase shift of exactly $\pi$ between Fig. 6a, b. This demonstrates the isolation of the sixth-order nonlinear signal from unwanted lower-order contributions such as cascade processes[76]. The phase shift results from the different prefactors of the perturbative expansion, i.e., $i^4 = 1$ versus $i^6 = -1$. This finding is also in excellent agreement with the simulations using identical parameters with the same model as above (see Supplementary Note 11 and Supplementary Fig. 13).

We further note that the rephasing and nonrephasing 2Q–0Q–1Q signals represent the direct population-based analogues of the coherently detected signals employed in the recently introduced exciton–exciton-interaction (EEI) 2D spectroscopy[31]. Thus, we deem that these sixth-order signals are also susceptible to exciton–exciton annihilation dynamics over $T$ for excitonic systems.

We introduced three-dimensional (3D) fluorescence-detected coherent spectroscopy and demonstrated the potential of the technique on a dianionic molecule in solution. By using shot-to-shot pulse shaping in a fully collinear single-beam setup, nearly half a million pulse trains were streamed in just 8 min. This time scale is

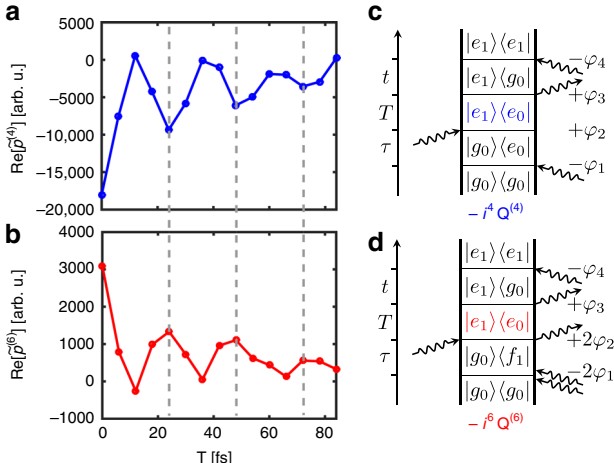

**Fig. 6** Comparison of fourth- and sixth-order 0Q coherence signatures of TIPS–TAP$^{2-}$ in THF. **a** Real part of a cross-peak transient of the rephasing 1Q–0Q–1Q contribution ($\tilde{p}^{(4)}$) at ($\hbar\omega_\tau$, $\hbar\omega_t$) = (2.06, 2.23 eV), along with **c** a respective double-sided Feynman diagram. **b** Real part of a cross-peak transient of the rephasing 2Q–0Q–1Q contribution ($\tilde{p}^{(6)}$) at ($\hbar\omega_\tau$, $\hbar\omega_t$) = (4.27, 2.23 eV) with **d** a respective double-sided Feynman diagram

short enough to capture rich information about a molecular system with limited chemical stability. A 125-fold phase-cycling procedure enabled us systematically and simultaneously to acquire multiple fourth- and sixth-order 3D spectra that were encoded into the monitored fluorescence signal, offering access to various kinds of high-order nonlinear signal contributions. From these, we obtained the characteristics of a two-photon allowed state, such as its energy, vibrational displacement, and highly efficient internal conversion to $S_1$. We also find a significantly longer dephasing time of 1Q coherences between the first and the two-photon excited state, reflecting a degree of correlation between the energy fluctuations of these two excited states, relative to the ground state. In addition, the simultaneous acquisition of rephasing and nonrephasing sixth-order signal contributions allowed us to obtain purely absorptive 2Q spectra from the same measurement. Our approach does not necessitate changing any physical setup component in order to

obtain a certain nonlinear signal contribution, and can readily be upscaled in terms of even further extended phase cycling and dimensionality. The possibility to acquire multiple-quantum signals in a systematic fashion by phase cycling underlines the enormous versatility of this technique which can also be employed for other incoherent detection modes such as the detection of electrons or ions[77,78]. From a total of 15 different signal contributions for the phase-cycling scheme illustrated herein, some of the nonlinear signals may prove useful, especially in excitonic systems, to decipher multiexcitonic coupling patterns, to determine bi- and triexciton binding energies, or to track selectively exciton–exciton annihilation, all from a single measurement.

## Methods

**Sample preparation and handling**. The sample is a 0.4 mM solution of the potassium salt of the dianion of 6,13-bis((triisopropylsilyl)ethynyl)quinoxalino[2,3-b]phenazine (TIPS–TAP$^{2-}$) in THF. The compound was synthesized by reducing the respective neutral species with two equivalents of [K(18-crown-6)(THF)$_2$] anthracenide, as previously described[52,53]. The preparation was conducted in an argon-filled glove box. THF was stored over NaK inside the glove box and filtered several times through alumina. After preparation, the dark-red solution of TIPS–TAP$^{2-}$ was transferred into a home-built sealed tubing system under inert conditions. During the multidimensional experiment, the sample was continuously pumped through a capillary flow cuvette with $(250 \, \mu m)^2$ cross-section (131.310-QS, Hellma) by a peristaltic pump (Masterflex L/S model 7518-00). Fluorescence spectra were recorded immediately before and after the multidimensional experiment, confirming that no possible decay products contaminated the nonlinear signals (see also Supplementary Note 1 and Supplementary Fig. 3).

**Experimental procedure and data processing**. The experimental setup for single-beam shot-to-shot multidimensional fluorescence spectroscopy has been reported elsewhere[55,65]. For a detailed schematic, see Supplementary Fig. 1. Briefly, an argon-filled (1.3 bar absolute pressure) hollow-core fiber was used in combination with an acousto-optical programmable dispersive filter (DAZZLER, Fastlite) to produce broadband phase-modulated four-pulse sequences with variable time delays and variable phases between all four pulses. We used a dual grism compressor for dispersion compensation. Collinear frequency-resolved optical gating (cFROG)[79] yielded a temporal pulse duration of < 21 fs (intensity FWHM), whereas the excitation energy is 200 nJ (determined at maximum temporal overlap and complete constructive interference of all four excitation pulses). In the multidimensional experiment we employed 125-fold phase cycling, allowing us to extract the various signal contributions out of the raw data by employing interpulse phase weights as described in the Results section. The excitation pulse trains were focused into the sample cuvette via a $f = -150$ mm focusing mirror. Fluorescence was collected perpendicular to the direction of the incoming laser beam by two 0.25-NA microscope objectives which were coupled to an avalanche photodiode (APD) via a 400-μm core-diameter multi-mode fiber. In order to avoid nonlinear contributions from the APD, the fluorescence signal was attenuated to a signal level within the linear detection regime via neutral density filters. To improve data quality, we averaged over four raw data sets. Each signal contribution was extracted by employing phase weights according to Eq. (1) and Table 1. In order to avoid truncation artifacts stemming from long-lived dynamics that evolve during $T$, the respective signal contribution was first apodized with a Hanning window. Before 3D Fourier transformation, the initial time-domain values of all temporal dimensions were scaled by a factor of 0.5 to suppress spurious peak-shape artifacts[80] and further three-fold zero-padded in every temporal dimension.

## Data availability

The data that support the findings of this study are available from the corresponding author upon reasonable request.

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

## Acknowledgements

We acknowledge funding by the European Research Council (ERC) within Consolidator Grant No. 614623, the Deutsche Forschungsgemeinschaft (DFG, German Research Foundation)—423942615 (T.B.), and the DFG with the University of Würzburg in the funding programme Open Access Publishing. T.B., T.B.M., and C.L. thank the Bavarian State Ministry of Science, Research, and the Arts for funding within the Collaborative Research Network "Solar Technologies Go Hybrid". We thank Walter Pfeiffer and Tristan Kenneweg for help with calculations and Christoph Brüning, Aleksander Paravac, and the Rechenzentrum of the University of Würzburg for providing computing resources. We also acknowledge Florian Hirsch concerning the design of the tubing system as well as Evripidis Michail for the two-photon absorption measurement, Mario Guth for help with the tubing system, and Zhu Wu for assistance during synthesis.

## Author contributions

S.M. proposed and designed the experiment, analyzed the data, and carried out simulations of the 3D spectra. S.M. and J.L. performed the experiment. S.M. and P.M. derived a theoretical model for simulating the 3D spectra. T.B. supervised the project. L.J. and J.L. synthesized the dianionic compound, supervised by T.B.M. and U.H.F.B., while J.H. performed TD–DFT calculations, supervised by A.D. Linear absorption was measured by M.M., supervised by C.L., via spectroelectrochemistry. S.M., J.L., P.M., and T.B. wrote the paper, with input from all coauthors. All authors contributed to the discussion of the results.

## Competing interests

The authors declare no competing interests.
