## [Peer Review File · Nature Communications]

Reviewers' comments:

Reviewer #1 (Remarks to the Author):

This paper presents a pulse-shaping fully collinear approach to acquiring multiple types of multidimensional spectroscopic signals using fluorescence detection. The study builds on numerous related advances using phase-cycling and phase modulation to perform third and higher order nonlinear spectroscopies in fully and partially noncollinear setups. The authors demonstrate their approach in TIPS-TAP2- and show good agreement between their data and simulations of the various spectroscopic signals. While the idea of a single measurement providing multiple difficult multidimensional spectroscopic measurements is appealing, the implementation has serious limitations that require further discussion as detailed below before they make a sufficiently convincing case for publication in Nature Communications.

1) In their implementation the authors use a single pulse-shaper to produce the required pulse sequence and with appropriate relative phases to perform their experiment. The authors should explicitly discuss the considerable limitations of their implementation in terms of ability to scan the t , T and τ delays, which will limit significantly the types of systems and questions that can be addressed with their method. For example, the authors cite the work of Turner and Nelson (ref 16) as motivation for using higher than third order measurements to study electronic correlations in semiconductors. In ref 16 the authors scan ~ 2.5 ps delays to for two-quantum coherence measurements. In ref 30, which the authors cite as motivation for using their approach to track multistep energy transfer in photosynthetic systems, one of the delays is scanned to 8ps. In general it will be desirable to scan long delays (~ 100 ps) for this type of application. The authors should provide details about what delays are typically necessary for various measurements on different systems and clearly describe the capabilities and limitations of their setup to perform these measurements.

2) Some of the advances over fully noncollinear measurements that are detailed in the paper have previously been shown for the partially collinear "pump-probe" geometry. This should be acknowledged in the paper. For example, the inherent phase stability, lack of need for "phasing the data" and use of phase-cycling for the separation of rephasing and nonrephasing signals have all been previously demonstrated.

3) The authors should cite the related work by Karki et al (Journal of Optics, 18 (2016) 015504) which shows that a very similar approach using phase-modulation can be used to simultaneously collect multiple coherent and incoherent nonlinear signals in a fully collinear geometry.

4) The 4th order spectroscopy performed by the authors purports to provide insight into the character of the electronic two-photon state of their sample. They should cite related previous work by Li et al (JCP 126, 164307, 2007) and comment on its relation to their study.

Reviewer #2 (Remarks to the Author):

Hereby review of Mueller et al., "Rapid multiple-quantum three-dimensional fluorescence spectroscopy disentangles quantum pathways." The authors demonstrate the power of fluorescence detected multi-dimensional spectroscopy by showing proof-of-principle multiple-quantum three-dimensional spectral bodies. The novelty speaks for recommending publishing. I have, however, a strong concern related to the sample degradation. The Supplementary Figure 2 clearly shows that the fluorescence has dropped by a factor of 4 during the measurement. Wouldn't this affect the outcome of the reconstruction by eq. (1)? Due to the degradation, the experimental data points p have additional coefficient which changes during the measurement from 1 at the beginning to 0.25 at the end. In the eq. (1) basically cancellation of all unwanted signals occurs leaving only the required signal. If the signal drops during

the measurement, wouldn't this delicate behavior of the eq. (1) be destroyed? Before this work can be published, the authors need to carefully explain why doesn't the sample degradation affect their analyses.

Other comments/questions:

1. In conventional coherently detected 2D certain higher order terms can contribute to the third order phase matched direction – one can always add $0 = k_1 - k_1$, for example. Is there an analogous possible distortion even here in case of fourth order possibly being “contaminated” by the 6-th order?

2. Eight minutes is with each shot being different. Here three sets were measured for averaging. How realistic is it that one can actually perform the measurement within 8 min?

3. In the linear absorption spectrum there seems to be another spectral feature (other mode) approximately half of the mode that the authors discuss. It's HR factor is smaller but the spectral coverage of it by the pulse would be much stronger. Why doesn't this show up in 2D experiment?

4. Before Fig 2: “first-excited 1Q, and doubly-excited 2Q states” is misleading. 1Q and 2Q refer to the coherences between the states rather than states themselves. There can be 1Q coherence even between e and f, e.g.

5. Pg 10.: “The overall amplitude of the 3D solid is determined by the convolution of $|e_0\rangle$ and $|e_1\rangle$ with the laser spectrum.” Should it be convolution or multiplication?

6. Pg 12 in the middle. The 2Q is the w_T dimension – check the 1Q-2Q frequency cut assignment, something seems to be out of order here.

7. Pg 17. “convoluting the homogeneous linewidth of 1Q and 2Q coherences into the spectral response” Do the authors mean convolution in the mathematical meaning of the word here? If not, I would change it to something else.

8. Pg 23 “Consolidator Grant Grant...”

9. Authors generally do a good job citing earlier relevant work. Maybe only the positive-negative w_T analyses, which is quite powerful, could deserve a few more early references, like DOI 10.1021/jp406103m and 10.1021/acs.jpcclett.7b00612

Reviewer #3 (Remarks to the Author):

Mueller et al demonstrate a novel implementation of coherent non-linear spectroscopy via fluorescence detection in a prototypical molecular system. A rapid yet broad phase-cycling scheme enabled the authors to obtain an impressive multi-dimensional dataset within a few minutes and such a scheme can indeed enable the use of non-linear spectroscopy on challenging yet relevant samples. The authors present various multi-quantum pathways in a fluorescent dianionic molecule, involving both vibrational and electronic coherences visualized in three-dimensional spectra. A few minor concerns are listed below:

1. While the SNR in the dataset looks impressive for such a quick measurement, the authors should clearly state the fluences employed in the measurements in the main text. Given that high excitation densities are not always preferred in order to reduce multi-particle interactions which may broaden the coherent spectra, it will be beneficial if the authors can discuss the fluence limits of their

experimental scheme. Also, recently Silva and co-workers (J Chem Phys 147, 114201) have highlighted possible contributions from incoherent photoluminescence non-linearities due to bimolecular recombination processes which may contribute to the PL detected non-linear spectra. Authors should comment on such incoherent contributions in the intensity regime employed in the current experiment.

2. Zero-quantum experiment:

(a) I suggest authors to highlight the A and C band in the 3D spectra more clearly in Figure 3.

Separation of these bands from the rest of the signal, as the authors did with the non-rephasing spectra in the supporting information would enable a better estimation of their relative energetics.

(b) The ability to distinguish ground state and excited state vibrational coherences is plus to the experiment. Can the authors comment on the differences or similarities of these coherences in the current dataset? It appears that the GS and Excited state vibrations have same energies at 0.17 eV from the what has been considered for simulations?

3. Two-quantum experiment:

(a) The existence of S1 to Sx transition should have manifested as a ESA feature, as also rightly pointed out by the authors in pages 14-15. Despite having a strong 2Q signature and finite oscillator strength for S1-Sx transition, the authors do not observe any ESA in their 1Q spectrum, why?

(b) The authors state - The solid shows further a vibrational shoulder along T with a higher magnitude compared to the 1Q progression band. This is not very apparent from the figure, and also can there be convolution from the pump spectrum?

4. For the sixth order experiments, the authors should discuss the possible contributions from multi-particle excitations rather than multi-quantum excitations by $n\phi$ pulses as recently presented by Mukamel (J. Chem. Phys. 145, 041102 (2016)).

Reply to Reviewer Comments on Nature Communications Manuscript NCOMMS-19-14399

Title: "Rapid multiple-quantum three-dimensional fluorescence spectroscopy disentangles quantum pathways"

Authors: Stefan Mueller, Julian Lüttig, Pavel Malý, Lei Ji, Jie Han, Michael Moos, Todd B. Marder, Uwe H. F. Bunz, Andreas Dreuw, Christoph Lambert, and Tobias Brixner

We want to thank all reviewers for their objective and detailed comments that helped us to improve the clarity of the presentation in the revised version of our work. We have addressed all points in detail as listed below, and we have modified the manuscript accordingly, including modification of some figures. We further included additional references as suggested by the reviewers.

Reviewers' comments in this reply letter are printed in black, our response in blue, and explicit changes to the manuscript in red font. All modifications are also visible in a marked-up version of the manuscript. Whenever page numbers are provided in our reply, they refer to the revised, marked-up version. If not stated otherwise, referenced numbers also refer to the revised, marked-up version.

Reviewer #1

This paper presents a pulse-shaping fully collinear approach to acquiring multiple types of multidimensional spectroscopic signals using fluorescence detection. The study builds on numerous related advances using phase-cycling and phase modulation to perform third and higher order nonlinear spectroscopies in fully and partially noncollinear setups. The authors demonstrate their approach in TIPS-TAP2- and show good agreement between their data and simulations of the various spectroscopic signals. While the idea of a single measurement providing multiple difficult multidimensional spectroscopic measurements is appealing, the implementation has serious limitations that require further discussion as detailed below before they make a sufficiently convincing case for publication in Nature Communications.

We have addressed all raised issues in our revised version as detailed below.

1) In their implementation the authors use a single pulse-shaper to produce the required pulse sequence and with appropriate relative phases to perform their experiment. The authors should explicitly discuss the considerable limitations of their implementation in terms of ability to scan the t , T and τ delays, which will limit significantly the types of systems and questions that can be addressed with their method. For example, the authors cite the work of Turner and Nelson (ref 16) as motivation for using higher than third order measurements to study electronic correlations in semiconductors. In ref 16 the authors scan ~ 2.5 ps delays to for two-quantum coherence measurements. In ref 30, which the authors cite as motivation for using their approach to track multistep energy transfer in photosynthetic systems, one of the delays is

scanned to 8ps. In general it will be desirable to scan long delays (~100ps) for this type of application. The authors should provide details about what delays are typically necessary for various measurements on different systems and clearly describe the capabilities and limitations of their setup to perform these measurements.

The main limitations of our experiment are given by the core device of the setup which is the acousto-optic programmable dispersive filter (AOPDF). In detail, the limitation in scanning delays, i.e., the temporal shaping window, is directly correlated with the length of the TeO₂ crystal inside the AOPDF. Our AOPDF model, the Dazzler WR510-950, supports a total temporal shaping window of 6 ps (@530 nm) to 8 ps (@900 nm) according to the manufacturer, so we can reach delays that are comparable to some of those quoted by the reviewer (e.g., 2.5 ps or 8 ps). Scanning even longer delays up to 100 ps would facilitate to measure complex energy transfer mechanisms or exciton diffusion dynamics. While this cannot be realized with our current setup, there exist nevertheless many feasible applications of our method to molecular systems that make up for the limitation in scanning range.

One particular strength of the fluorescence-based method is the absence of coherent artifacts at early population times compared to many coherently-detected setups that have significantly larger delay scan capabilities. This underlines that our presented method is explicitly advantageous for early delay times, making it already greatly suitable to study ultrafast processes like rapid exciton–exciton annihilation in many classes of molecular aggregates.

Anyway, no single method is likely to be optimal in all technical parameters, and we do not claim that this was the case for our method. Existing approaches all have their merits.

We added a discussion of the temporal scanning limitations on p. 9: “Our technique is currently limited to a maximum temporal shaping window of 8 ps for every pulse sequence according to the length of the chosen AOPDF crystal. At such delays, one can measure electronic and vibrational coherence dynamics of dissolved molecular systems that proceed on time scales up to several picoseconds. In cases of much slower dephasing, such as in some semiconductor nanostructures^{15,16,59}, measurements of all relevant dynamics becomes challenging. Another example of slow dynamics proceeding at the 10 to 100 ps time scale is exciton diffusion dynamics in some molecular aggregates³¹. Nevertheless, because of the absence of spurious coherent artifacts for early population times, our technique has great application potential to study ultrafast processes such as rapid exciton–exciton annihilation in many classes of molecular aggregates, and the single-beam geometry provides a compact, stable, and convenient access.”

2) Some of the advances over fully noncollinear measurements that are detailed in the paper have previously been shown for the partially collinear “pump-probe” geometry. This should be acknowledged in the paper. For example, the inherent phase stability, lack of need for “phasing the data” and use of phase-cycling for the separation of rephasing and nonrephasing signals have all be previously demonstrated.

We have included a statement accordingly on p. 3 in the manuscript: “In this context, it was previously shown that a partially collinear pump–probe geometry offers many advantages due to its inherent phase stability, and the capability to isolate several, directly-phased signals at once^{33–37}.”

3) The authors should cite the related work by Karki et al (Journal of Optics, 18 (2016) 015504) which shows that a very similar approach using phase-modulation can be used to simultaneously collect multiple coherent and incoherent nonlinear signals in a fully collinear geometry.

We included this reference into the manuscript on p. 3, where we write “As another alternative that also preserves these advantages, one can employ phase modulation or phase cycling with fluorescence detection^{38–47}.”

4) The 4th order spectroscopy performed by the authors purports to provide insight into the character of the electronic two-photon state of their sample. They should cite related previous work by Li et al (JCP 126, 164307, 2007) and comment on its relation to their study.

We refer to the relevant work by Li et al on p. 9 in the manuscript, where we now write:

“... and 2Q coherences between ground- and higher-excited electronic states via multiphoton processes. Li *et al.* previously introduced multiphoton 2D fluorescence spectroscopy, where signatures of one- and two-photon transitions into the same electronic state could be obtained⁵⁸. In the present work, however, we will consider one- and two-photon transitions into different electronic states.”

Whereas an experiment in the same fashion as in the work by Li *et al.* could also be designed for our case, it would be rather less useful, because in TIPS-TAP²⁻, all one-photon transitions are two-photon forbidden and vice versa because of its inversion symmetry.

Reviewer #2:

Hereby review of Mueller et al., “Rapid multiple-quantum three-dimensional fluorescence spectroscopy disentangles quantum pathways.” The authors demonstrate the power of fluorescence detected multi-dimensional spectroscopy by showing proof-of-principle multiple-quantum three-dimensional spectral bodies. The novelty speaks for recommending publishing. I have, however, a strong concern related to the sample degradation. The Supplementary Figure 2 clearly shows that the fluorescence has dropped by a factor of 4 during the measurement. Wouldn't this affect the outcome of the reconstruction by eq. (1)? Due to the degradation, the experimental data points p have additional coefficient which changes during the measurement from 1 at the beginning to 0.25 at the end. In the eq. (1) basically cancellation of all unwanted signals occurs leaving only the required signal. If the signal drops during the measurement, wouldn't this delicate behavior of the eq. (1) be destroyed? Before this work can be published, the authors need to carefully explain why doesn't the sample degradation affect their analyses.

We thank the reviewer for the positive recommendation and this constructive comment. To address the degradation problem, we added several new paragraphs to Supplementary Note 1 and also created new Supplementary Figure 4, for convenience copied below. In Supplementary Note 1, we now write on p. 4:

“The question arises if the sample degradation would affect the reconstruction of nonlinear signals. We will now discuss why our sampling scheme avoids to first order those errors that might be connected with sample decay.

The most crucial criterion for extracting nonlinear contributions according to their phase signatures by Eq. (1) in the manuscript is that the signal level should not drop dramatically within the phase-cycling procedure for any particular given setting of time delays. Hence, it would be detrimental if we had sampled the phase-cycling steps after scanning the complete manifold of interpulse time delays. In our sampling scheme, however, we first acquire all the phase-cycling steps for any given combination of interpulse delays before we change the latter to different settings. Due to the availability of shot-to-shot pulse-shape modulation, acquiring a 125-step phase-cycling set takes 125 ms at 1 kHz repetition rate. This time scale is significantly smaller than the time scale on which the sample degrades, so that the phase-cycling procedure is not compromised by sample degradation.

We characterize the sample degradation during the measurement by recording several “reference” data points, consisting of a repetition (for averaging) of a single, compressed, but otherwise unshaped laser pulse. This reference measurement corresponds effectively to setting all interpulse time delays and interpulse phase differences to zero. These reference data are acquired at the beginning and the end of the full dataset as well as before any population time increment. Thus, between two averaged reference data measurements, the 125-step phase cycling is conducted for all delay increments of τ and t at a given, fixed T . The resulting reference data points are shown in Supplementary Figure 4.

Supplementary Figure 4. Time-integrated fluorescence signals that were acquired as reference data (by averaging over a repetition of a single laser pulse) during the multidimensional experiment. Between any two reference points, 125 different phase-cycled data points for each τ and t delay are acquired (not shown).

In our specific case, analyzing the decay of the reference measurements, we find that the signal changes on average by 3.5% between two reference measurements, so that the claim of constant intensity for any given phase-cycling set is fulfilled up to that precision throughout all τ and t combinations for any T , and the intensity decreases by less than 0.02% for a 125-step phase cycling protocol at any fixed τ and t combination.

The signal decays for instable samples throughout the sampling of T steps because these are sampled last. The degradation acts then as an additional damping of the actual dephasing dynamics of coherences that evolve during T . This has most effects on the 0Q coherence dynamics which usually dephase within several picoseconds. The sample decay then manifests as an additional broadening of the 3D line shapes along the ω_T axis in the 3D spectra. However, a broadening along ω_T for 0Q coherences is introduced anyway by the Hann window we applied in order to avoid truncation artifacts from Fourier transform. Hence, this additional broadening by sample decay might be present but it does not change the spectral position of 0Q features.

Finally, we point out that a sample decay over the course of the acquisition time is not particular in any way to the present technique, but would rather occur with any other approach. On the contrary, using the present shot-to-shot modulation technique, the effect of sample degradation can be reduced as far as possible and thus facilitates measurement of unstable species that are not amenable to analysis with conventional means.”

We refer to the discussion above in the manuscript on p. 8 (bottom): “In our sampling protocol, we first sample through all phase-cycling steps for fixed interpulse delays to prevent errors in reconstructing the phase-sensitive nonlinear signal contributions by Eq. (1) (for further details see Supplementary Note 1).”

Other comments/questions:

1. In conventional coherently detected 2D certain higher order terms can contribute to the third order phase matched direction – one can always add $0 = k_1 - k_1$, for example. Is there an analogous possible distortion even here in case of fourth order possibly being “contaminated” by the 6-th order?

We did not address this specifically in the manuscript, but indeed the situation is identical to the coherently-detected case. Hence, we added in the manuscript on p. 8 (middle): “However, analogously to coherently-detected experiments³¹, if the excitation density is high enough to generate sixth-order signals, these may also contaminate the fourth-order ones. Hence, whenever we use the term “fourth-order signal”, we mean a nonlinear signal that is predominantly of fourth order (see Supplementary Note 3). Apart from that, an advantage of our technique is that we are able to simultaneously isolate the higher-order contribution by appropriate phase-cycling.”

In new Supplementary Note 3, we provide additional discussion on that issue: “A common fundamental assumption in third-order nonlinear spectroscopic experiments employing coherent detection is that the third-order signals dominate over the higher nonlinear orders (e.g., fifth-order signals). However, once fifth-order signals are present, these contaminate the third-order signals. This results from the emission of the signal of both nonlinear orders into the same phase matched direction, because one can in principle add zero to the wave vector of any of the incident beams (e.g., $0 = k_1 - k_1$). Due to the similar nature of phase matching and phase cycling, this effect is also expected in our experiment, meaning that for any pulse, interaction phases of, e.g., $0 = \varphi_1 - \varphi_1$ can be added. As a result, fourth-order signals can be contaminated by sixth-order signals, analogously to the case that third-order signals can be contaminated by fifth-order ones. We have discussed the latter previously in the context of our development of exciton–exciton-interaction 2D (EEI2D) spectroscopy [Dostál et al., *Nat. Commun.* 9, 2466 (2018)]. In that work, we showed that the contamination of third-order signals, due to exciton–exciton annihilation, for example, can be characterized by separately and simultaneously measuring a fifth-order EEI2D signal.

In the present work, we compare the absolute-valued signal magnitudes of the fourth-order rephasing 1Q-0Q-1Q and sixth-order rephasing 2Q-0Q-1Q signals that we also simultaneously measure and find that the sixth-order signal has only 4.96% of the magnitude of the fourth-order one. Hence, because the sixth-order signal is over one magnitude weaker than the fourth-order one, any possible distortion would be at the 5% level or lower.

Considering all the evidence enables us to assign the nominally labeled “fourth-order spectra” indeed to result predominantly from the fourth-order nonlinear response.”

2. Eight minutes is with each shot being different. Here three sets were measured for averaging. How realistic is it that one can actually perform the measurement within 8 min?

We implemented new Supplementary Figure 2 (for convenience plotted below) and added a brief discussion about the data quality that can be achieved without any averaging into Supplementary Note 1 (p. 3): “The data presented in the manuscript were averaged over four complete datasets. In Supplementary Figure 2, we show exemplarily the fourth-order 1Q-0Q-1Q and the sixth-order 2Q-0Q-1Q 3D spectra resulting from a single data set without averaging. The same plotting parameters were used as those described in the captions of manuscript Figs. 3 and 5, respectively.

Supplementary Figure 2: **a** Rephasing 1Q-0Q-1Q and **b** 2Q-0Q-1Q 3D spectra after a single measurement without averaging.

It is evident that an excellent signal-to-noise ratio is achieved for the fourth-order 3D spectrum after only a single run, i.e., without any averaging. For the sixth-order contribution that is much weaker, the noise is significantly higher compared to the four-times averaged data in Fig. 5 of the manuscript. Nevertheless, relevant signal features are already visible. This shows that it is indeed possible to obtain all fourth-order signals in 8 min, while for the sixth-order signals additional averaging is helpful.”

3. In the linear absorption spectrum there seems to be another spectral feature (other mode) approximately half of the mode that the authors discuss. It’s HR factor is smaller but the spectral coverage of it by the pulse would be much stronger. Why doesn’t this show up in 2D experiment?

As correctly pointed out by the reviewer, the spectral coverage by the employed laser spectrum is much larger in the region of this lower-frequency vibration, which has an energy of ≈ 81 meV (≈ 653 cm^{-1}) and is a pronounced feature in the linear absorption spectrum. However, throughout all nonlinear spectroscopies, the total signal in the frequency domain is always given by a product of the system’s response with the spectrum of the external electric fields, which we also need to consider here. In this respect, the product of the dominating main band of the S_0 - S_1 0-0 transition with the laser spectrum might superimpose possible features from the lower-frequency mode due to “laser pulling” effects. In order to illustrate this, we show the

product of the linear absorption spectrum of the molecule with the measured laser spectrum in Fig. R3 below.

Figure R3: Linear absorption spectrum (black) of TIPS-TAP²⁻ in THF together with an overlay of the product spectrum (red) of linear absorption with the laser spectrum that is employed for 3D spectroscopy. Each spectrum is normalized to its maximal value.

While in the linear absorption spectrum all vibrational progression bands appear well-separated (Fig. R3, black), in the product spectrum (Fig. R3, red) the vibrational mode at ~ 2.14 eV appears only as a shoulder of the main band and not as a separate peak. Thus, we do not see a distinct feature of it in the 3D experiments.

In addition, we think it is difficult to see quantum beatings of the 81 meV mode because of (1) the quite low oscillator strength of this vibration and (2) the relatively high oscillation period (≈ 51 fs) compared to our sampling window for T . The only quantum beatings we observe are those of the ≈ 170 meV mode, where we could sample three oscillation periods.

4. Before Fig 2: “first-excited 1Q, and doubly-excited 2Q states” is misleading. 1Q and 2Q refer to the coherences between the states rather than states themselves. There can be 1Q coherence even between e and f, e.g.

While these phrases can commonly be found in 2D literature, we agree that they are misleading. We discarded the additional “1Q” and “2Q” terms on p. 10 and further changed on p. 14, middle: “While 1Q-0Q-1Q spectra give extensive information about the manifold of singly-excited states...”. We also revised similar statements on p. 16, top: “...so that one can probe how these states are coupled to the manifold of singly-excited states.”, and on p. 20, top: “We observe features at positions A and C signifying that $|e_0\rangle$ and $|e_1\rangle$ are coupled to the lowest level of the doubly-excited manifold, $|f_0\rangle$, whereas features B and D indicate that the singly-excited manifold is coupled to $|f_1\rangle$ as well.”

5. Pg 10.: “The overall amplitude of the 3D solid is determined by the convolution of $|e_0\rangle$ and $|e_1\rangle$ with the laser spectrum.” Should it be convolution or multiplication?

Indeed, the reviewer is correct, in the frequency domain it is a multiplication (and in the time domain a convolution). We changed accordingly “convolution” into “multiplication” on p. 11 in the manuscript.

6. Pg 12 in the middle. The 2Q is the w_T dimension – check the 1Q-2Q frequency cut assignment, something seems to be out of order here.

We checked the entire p. 12 in the original manuscript for errors regarding the clauses that the reviewer mentions, but could not find anything that fits this concern. We assume that the reviewer meant something else. Thus, we also checked all plots depicted in Fig. 4 which display the 1Q-2Q projections for correctness and could not find any error in these representations either.

7. Pg 17. “convoluting the homogeneous linewidth of 1Q and 2Q coherences into the spectral response” Do the authors mean convolution in the mathematical meaning of the word here? If not, I would change it to something else.

The term was not applied in its correct mathematical meaning. Thus, we changed this clause into “..., thus the homogeneous linewidth of both 1Q and 2Q coherences contribute to the spectral response.”

8. Pg 23 “Consolidator Grant Grant...”

We corrected this typo.

9. Authors generally do a good job citing earlier relevant work. Maybe only the positive-negative w_T analyses, which is quite powerful, could deserve a few more early references, like DOI 10.1021/jp406103m and 10.1021/acs.jpcllett.7b00612

We thank the reviewer for these suggestions and included these references on p. 13, where we added the following: “As previously demonstrated, analyzing these oppositely-signed signatures has proven to be a powerful tool to decipher the origin of the underlying quantum coherences^{62,63}.”

Reviewer #3:

Mueller et al demonstrate a novel implementation of coherent non-linear spectroscopy via fluorescence detection in a prototypical molecular system. A rapid yet broad phase-cycling scheme enabled the authors to obtain an impressive multi-dimensional dataset within a few minutes and such a scheme can indeed enable the use of non-linear spectroscopy on challenging yet relevant samples. The authors present various multi-quantum pathways in a fluorescent dianionic molecule, involving both vibrational and electronic coherences visualized in three-dimensional spectra. A few minor concerns are listed below:

We thank the reviewer for this positive comment and also for all the helpful suggestions below.

1. While the SNR in the dataset looks impressive for such a quick measurement, the authors should clearly state the fluences employed in the measurements in the main text. Given that high excitation densities are not always preferred in order to reduce multi-particle interactions which may broaden the coherent spectra, it will be beneficial if the authors can discuss the fluence limits of their experimental scheme. Also, recently Silva and co-workers (J Chem Phys 147, 114201) have highlighted possible contributions from incoherent photoluminescence non-linearities due to bimolecular recombination processes which may contribute to the PL detected non-linear spectra. Authors should comment on such incoherent contributions in the intensity regime employed in the current experiment.

We now state the fluence on page 20, where we also discuss why these incoherent contributions are unlikely in our experiment: “Another source for unwanted signals is so-called incoherent population mixing, that is, signals arising from interacting excited-state populations, as discussed by Grégoire *et al.*, which was observed in solid-state semiconductors under high excitation densities⁷³. In our experiment, we determined the number of absorbed photons per molecule³¹ to be 0.26, which would in principle facilitate the generation of such incoherent signals by bimolecular processes. Considering the relevance of this value for multi-particle interactions, however, we have to take into account the relative distance of chromophores because for systems that do not form aggregates (such as in the present case), the probability of bimolecular processes is generally low. According to the suggestion of Tiwari *et al.*, we have estimated the effect of multi-particle interactions for the present sample using Förster resonance energy transfer (FRET) as a crucial step that would facilitate the interaction of populations of two distinct molecules by subsequent exciton–exciton annihilation⁴³. We calculated the Förster critical concentration^{74,75} for this effect to be 0.54 mM which is above our concentration (0.40 mM) so that we expect such contributions in our experiment not to be present.”

In J-aggregates, polymers, or solid-state materials, the probability of bimolecular processes is much higher. In fact, we have recently developed the method of exciton–exciton-interaction two-dimensional (EEI2D) spectroscopy to make visible such effects in order to learn about exciton diffusion. We point out that the sixth-order contributions introduced in the present

work may deliver the identical EEI2D information in applicable systems, such that one can probe multi-particle interactions exclusively and on purpose.

Concerning the terminology of “incoherent photoluminescence non-linearities” used by the reviewer, we note that the employed classification of this effect depends on the basis set in which one describes the measurement. If we employ a full basis set containing all relevant chromophores (i.e., including the two potential interacting ones), then all observable signals arise naturally in a fully coherent fashion when considering all the Feynman diagrams. There are no additional “incoherent non-linearities”. If, on the other hand, one limits the description to an effective single-particle picture (for example, because a full description is not tractable due to the system size), then such terms arise because the Feynman diagrams for the single particle do not capture the full system interactions. We have discussed these relations also in our work on EEI2D spectroscopy [Dostál et al., *Nat. Commun.* 9, 2466 (2018)].

In our theoretical modelling of the current work, we considered a single-particle model without any treatment of collective effects. Hence, since the theory agrees in an excellent fashion with experiment, we conclude multi-particle interactions to be absent (in line with the estimate from the critical Förster concentration).

2. Zero-quantum experiment:

(a) I suggest authors to highlight the A and C band in the 3D spectra more clearly in Figure 3. Separation of these bands from the rest of the signal, as the authors did with the non-rephasing spectra in the supporting information would enable a better estimation of their relative energetics.

We revised Figure 3 accordingly and now include close-ups of the isolated 3D peaks A and B (we assume the reviewer actually meant band B, not C, because the latter label is not present in Figure 3). Hence, we added to the Figure caption: “The spectral volumes inside the red and blue cuboids, which are centered around peaks A and B are isolated and shown on the right. There, isosurfaces are drawn at 50, 70, and 90% of the isolated maximal signal amplitudes whereas the colorbars indicate the signal levels (normalized to the highest absolute value of the respective full 3D spectrum) for each contour line.”

From these new representations, we determined the coordinates of the maximum-valued voxels inside these peaks. For peak A, the 0Q energy at the maximum is +188 meV, whereas the maximum of peak B is located at –172 meV. We therefore added several sentences to the manuscript on p. 13, top: “In order to separate them from the dominating peaks, we isolate the voxels within the spectral volumes that span the regions in the vicinity of peak A and B, which is indicated by red and blue cuboids, respectively (Fig. 3a, right). By determining the coordinates of the maximum amplitude of each isolated cross peak, we find that these are shifted along the $\Delta\hbar\omega_T$ axis by +188 meV (A) and –172 meV (B). In view of the effective spectral resolution

(15.7 meV), these values are in fair agreement with the energy spacing $\hbar\omega_{0Q} \approx \pm 0.17$ eV between vibrational levels within S_0 and S_1 ...".

Because we now also show close-ups of peaks A and B in experiment and simulation, we added in the top paragraph of p. 14: "...the experimental results (Fig. 3a) can be very well reproduced by simulations (Fig. 3c) using Lindblad theory⁶⁴, in terms of peak positions as well as their shapes and amplitudes, even for the peaks A and B ...". We further added the prefix " \approx " to the 170 meV value every time when referring to this particular vibration (p. 13, middle; p. 13 bottom) in order to reflect that his value was also given as an estimate in Ji *et al.*, *J. Am. Chem. Soc.* 139, 15968-15976 (2017).

(b) The ability to distinguish ground state and excited state vibrational coherences is plus to the experiment. Can the authors comment on the differences or similarities of these coherences in the current dataset? It appears that the GS and Excited state vibrations have same energies at 0.17 eV from the what has been considered for simulations?

In our reply to comment 2(a), we quantified the 0Q energies of peaks A and B of Fig. 3 in the manuscript to be +188 meV and -172 meV, respectively. Because peak A is directly assigned to the excited-state vibration, one might attribute this difference in absolute energies to a hardening of the corresponding vibrational mode upon excitation into the electronic state S_1 . However, this reasoning is weakened by the fact that "although two pathways could contribute to peak B in principle, its amplitude is nearly identical to peak A (Fig. 3a, right)", as we write at the bottom of p. 13. This means that, because peak B is attributed to both excited- and ground-state vibrational coherence, contributions from ground-state vibrational coherence are strongly diminished. Hence a comparison between these peaks is less meaningful because both peaks predominantly bear signatures from excited-state vibrational coherence.

Concerning the differences in 0Q energies of peaks A and B, one should also bear in mind the effective spectral resolution (15.7 meV) in the present experiment. Hence, because the difference between the 0Q energies for peaks A and B (16 meV) is on the order of the effective spectral resolution, care must be taken in terms of interpretation. In the simulation, we explicitly assumed equal vibrational frequencies in S_0 and S_1 (0.17 eV). Both peaks appear at 0Q energies of 188 meV in the simulation. We believe that the small deviation of 0Q peak energies in the 3D spectrum from the value used as input in the simulation can be attributed to the effective spectral resolution. In principle, a higher level of confidence in determining 0Q energies directly from the 3D plots can be reached by scanning the population time T up to several picoseconds.

3. Two-quantum experiment:

(a) The existence of S_1 to S_x transition should have manifested as a ESA feature, as also rightly pointed out by the authors in pages 14-15. Despite having a strong 2Q signature and finite

oscillator strength for S1-Sx transition, the authors do not observe any ESA in their 1Q spectrum, why?

We write, at the bottom of p. 14 in the manuscript, that “we neglected any ESA signatures that could potentially contribute to the 1Q-0Q-1Q signals.” The reason for this, which we failed to point out, is that in population-based approaches, each ESA pathway has a counterpart with opposite sign. If internal conversion is efficient just as in the case of TIPS-TAP²⁻, then these two pathways exactly cancel which leads to a disappearance of ESA features. A similar situation is evident in the 1Q-2Q-1Q pathways of Fig. 4b of the manuscript, where pathways Q_B and Q_C cancel because of this internal conversion. To make this clearer, we incorporated a section about 1Q-0Q-1Q pathways in Supplementary Note 4 along with new Supplementary Figure 6 and added “..., but the 1Q-0Q-1Q ESA pathways cancel in pairs, as also discussed in Supplementary Note 4” to the end of p. 14 in the manuscript.

(b) The authors state - The solid shows further a vibrational shoulder along T with a higher magnitude compared to the 1Q progression band. This is not very apparent from the figure, and also can there be convolution from the pump spectrum?

For the 1Q-2Q-1Q 3D spectrum, our intention was to highlight the energy of the lowest level of the two-photon state S_x and the equality of the 1Q-2Q and 2Q-1Q 2D projections of the solid. Thus, regarding the viewing angle we used, we agree that this might be disadvantageous to recognize the progression band of S_x. Nevertheless, this progression band can be inspected from another perspective with better visibility in another 3D spectrum along the 2Q axis, which is the sixth-order 2Q-0Q-1Q spectrum in Fig. 5 of the manuscript. There, it is highlighted as feature “B”. Indeed, the progression band of S_x appears only as a shoulder. As correctly pointed out by the reviewer, this is due to the multiplication with the laser spectrum as discussed above. In the 2Q domain, the laser spectrum has its maximum at 4.20 eV (according to frequency doubling of the laser spectrum depicted in Fig. 2 of the manuscript), which is located between the v=0 (4.10 eV) and v=1 (4.27 eV) levels in the S_x state and in turn leads to the appearance of this shoulder.

We thus added on p. 16 (top) in the manuscript: “The solid shows further a vibrational shoulder along $\hbar\omega_T$ with a higher magnitude compared to the 1Q progression band, resulting from the multiplication with the laser spectrum which has its maximum at ≈ 4.20 eV in the 2Q domain.”

4. For the sixth order experiments, the authors should discuss the possible contributions from multi-particle excitations rather than multi-quantum excitations by $n\phi$ pulses as recently presented by Mukamel (J. Chem. Phys. 145, 041102 (2016)).

We thank the reviewer for the important advice to consider this relevant publication. Because our response to this issue necessitates a rather long and detailed discussion, we implemented new Supplementary Note 9 and Supplementary Figure 11 and refer to them on p. 20 (top) in the

manuscript, where we also state the conclusion of this discussion: “Additional unwanted signals, such as those originating from many-particle excitations of non-interacting molecules⁷² are not present in our experiment (for a detailed discussion see Supplementary Note 9).”

Further changes

Besides changes that were associated with reviewer comments, further minor changes have been implemented to the revised version of the manuscript:

- We revised the **arrows that represent excitation between electronic states in terms of vertical transitions** in Figure 2.
- In Figure 6, we previously wrongly labeled of the nonlinear signal contribution by using the symbol p . By taking into account the complex-valued quantity of these signals, we corrected this into " **\tilde{p}** ", also in the Figure caption.
- P. 12, in the caption of Figure 3: "...the 2D projections of the spectral solids..."
- P. 20 at the bottom: "...where we compare transients over T of the **rephasing** fourth-order 1Q-0Q-1Q (Fig. 6a)"
- On P. 13 at the bottom we now write: "...the amount of thermal energy that was present at the experiment (≈ 26 meV)...", where the " \approx " shall reflect that slight deviations in temperature are generally possible.
- On P. 18, we **moved references 70 and 71 from the end of the sentence to the endmost comma.**
- P. 23: We corrected the last sentence in the Sample preparation and handling section to "Fluorescence spectra were recorded **immediately** before and after the multidimensional experiment..." as it was also formulated in the correct way on p. 3 in the Supplementary Information.
- We changed P. 24, top to "...whereas the **excitation energy is 200 nJ (determined at maximum temporal overlap and complete constructive interference of all four excitation pulses)**" in order to avoid misconception in the specification of the excitation energy.
- P. 25 in the Acknowledgements: changed "BR2123/13 1" to "**423942615**" which is the actual project number of this funding agency that was made available in the meantime.
- We revised Refs. 47 and 49 according to their publication *Chem. Sci.* and in *Opt. Lett.*, respectively, that occurred in the meantime.
- We changed all headlines into lower case.

Changes to Supplementary Information:

- We **added page numbers.**
- We revised Supplementary Figure 5. In order to make clearer that two interactions from a single laser pulse by phase $\pm 2\varphi_n$ take place at the same time, **we aligned the respective two wavy arrows in a way that they now appear symmetrically around the vertical lines in the double-sided Feynman diagrams.**
- We corrected **Eq. 7 for missing brackets.**
- We changed the term "**convolution**" to "**multiplication**" at the bottom of p. 14.

REVIEWERS' COMMENTS:

Reviewer #1 (Remarks to the Author):

The authors have done a thorough job of addressing the various concerns of the reviewers. The manuscript is now suitable for publication in Nature Communications.

Reviewer #2 (Remarks to the Author):

The authors have answered all my questions and I think that the manuscript can be published as it is.

Reviewer #3 (Remarks to the Author):

The authors have addressed all my comments and the manuscript can be published.

Reply to Reviewer Comments on Nature Communications Manuscript NCOMMS-19-14399A

Title: “Rapid multiple-quantum three-dimensional fluorescence spectroscopy disentangles quantum pathways”

Authors: Stefan Mueller, Julian Lüttig, Pavel Malý, Lei Ji, Jie Han, Michael Moos, Todd B. Marder, Uwe H. F. Bunz, Andreas Dreuw, Christoph Lambert, and Tobias Brixner

We again thank the reviewers for their positive evaluation of our work and all recommending publication in the present form. No further changes were requested. We included an additional statement in the Acknowledgements section that was requested from a funding agency in the meantime and corrected further a few minor typos. Besides that, all modifications in the manuscript (see marked-up version) are due to editorial stylistic requests.

Reviewer #1

The authors have done a thorough job of addressing the various concerns of the reviewers. The manuscript is now suitable for publication in Nature Communications.

Reviewer #2

The authors have answered all my questions and I think that the manuscript can be published as it is.

Reviewer #3

The authors have addressed all my comments and the manuscript can be published.